# The Gut Microbiome in Psoriasis and Crohn’s Disease: Is Its Perturbation a Common Denominator for Their Pathogenesis?

**DOI:** 10.3390/vaccines10020244

**Published:** 2022-02-05

**Authors:** Maria Antonia De Francesco, Arnaldo Caruso

**Affiliations:** Institute of Microbiology, Department of Molecular and Translational Medicine, University of Brescia-Spedali Civili, P. le Spedali Civili, 1, 25123 Brescia, Italy; arnaldo.caruso@unibs.it

**Keywords:** microbiota, inflammation, gut, skin

## Abstract

Psoriasis and inflammatory bowel disease (IBD), including ulcerative colitis (UC) and Crohn’s disease (CD), are interlinked. In fact, the prevalence of IBD is higher in patients with psoriasis, with a risk of ulcerative colitis of 1.6-times higher than in the general population. Analogously, patients with psoriasis have a greater risk of developing IBD. Furthermore, they share some clinical features and pathogenic mechanisms. Both are chronic inflammatory diseases with a relapsing-remitting condition that persists for the patient’s whole life and exhibit increased permeability of the mucosal barrier of skin and gut, allowing an increased interaction of pathogens with inflammatory receptors of the immune cells. A key element in the pathogenesis of these diseases is represented by the microbiota; in particular, the gut microbiota is an important driver of CD pathogenesis, while in psoriasis changes in gut and skin microbiota have been described without a defined pathogenic function. Furthermore, genetic predispositions or environmental factors contribute to disease manifestation, with a central role attributed to the immune responses and, in particular, to a dysregulated role played by T helper 17 cells both in psoriasis and IBD. The purpose of this review was to summarize present information about the links between psoriasis, inflammatory bowel disease, in particular Crohn’s disease, and changes in gut and/or skin microbiome.

## 1. Introduction

Inflammatory bowel disease refers to two forms of idiopathic intestinal disease that differ in their location and degree of gut wall involvement: ulcerative colitis (UC) and Crohn’s disease (CD). They share some clinical features with psoriasis.

They are, in fact, chronic and inflammatory conditions, and their natural story is characterized by alternating periods of remission and flares.

Different studies evidenced that patients affected by psoriasis, psoriatic arthritis and ankylosing spondylitis have a higher incidence of IBD [1,2,3]. Patients with psoriasis have 1.6 times more risk of developing ulcerative colitis than healthy subjects [4]. On the other hand, 9.6% of patients with Crohn’s disease were more frequently associated with psoriasis compared to 2% of control individuals [5].

Psoriasis affects about 2–3% of the world population, leading to hyperproliferation of keratinocytes and to a decreased barrier function of the skin, with infiltration of activated inflammatory cells [6,7]. Ulcerative colitis (UC) and Crohn’s disease are characterized by inflammatory processes, limited in UC to the mucosa and submucosal layers of the colon, and in CD, to any site of the gastrointestinal tract with segmentary distribution of the lesions [8]. The pathogenesis of these diseases is associated with an altered immune response against gut microbiota (IBD) or skin (psoriasis) in genetically predisposed individuals after environmental triggers.

The overall incidence of psoriasis [9] and CD [10] is similar in the male and female population, opposite to what occurs in some other autoimmune diseases found more often in women, such as multiple sclerosis, systemic lupus erythematosus and Sjögren’s syndrome [11]. Geographic variation was observed in both diseases, with higher incidence rates in Northern Europe and North America compared to those found in most regions of Africa, Asia and the Middle East [12,13,14].

For IBD risk, some environmental factors were identified, such as the order of birth; it was, in fact, found that higher birth rank (> or =3) was significantly associated with a lower risk of IBD (odds ratio 0.68) [15]. Among the other factors involved, there were smoking [16], breastfeeding [17] and antibiotic administration [18]. On the contrary, for psoriasis, the role of environmental factors is less evident, while it is known that stress and infections may act as stimuli of the disease [19].

Although the environment plays an important role, it is equally recognized that a genetic susceptibility is an underlying significant risk for both psoriasis and IBD. Genome-wide association studies have already reported the presence of 13 psoriasis susceptibility loci (denominated PSORS1-PSORS13), 32 loci associated with CD, and 17 associated with UC [3,20].

These studies also identified some functions of psoriasis susceptibility genes such as antigen presentation, specific cytokines and their cytokine receptors, downstream inflammatory signaling pathways, and epithelial functions [21,22,23,24]; similarly, they found new IBD pathways encompassing both innate and cell-mediated immunity [25,26].

A large meta-analysis including >4500 patients and 10,000 controls identified 11 susceptibility loci that were shared by IBD and psoriasis. Of these, seven were outside the human leukocyte antigen region. They included 9p24 near JAK2, 10q22 at ZMIZ1, 11q13 near PRDX5, 16p13 near SOCS1, 17q21 at STAT3, 19p13 near FUT2, and 22q11 at YDJC. Additionally, there were four loci that were already commonly known as strongly associated with both psoriasis and CD (IL23R, IL12B, REL, and TYK2) [27].

Of note, between the overlapping loci, there were ZMZ1, which encodes for protein zinc finger MIZ type 1, responsible for the activation of different transcription factors, and the gene suppressor of cytokine signaling 1 (SOCS1), a gene that mediates the negative regulation of cytokines activated by the JAK/STAT pathway.

CD and psoriasis are associated with different comorbidities [28]. Arthropathy is present in both diseases, but psoriasis is highly associated also with other pathologic conditions such as obesity, diabetes, hypertension, cardiovascular disease, malignancies, dementia, nonalcoholic fatty liver disease, and kidney disease [29,30,31]. Patients with CD may have some extraintestinal inflammation such as sclerosing cholangitis, ocular inflammation, and skin inflammation [32,33].

There is an overlapping modality of treatment for both diseases, even if with some differences. In psoriasis, the first-line treatment is based on phototherapy, retinoids, cyclosporine and methotrexate.

In the case of therapeutic failure, different biologic treatments are used, which are able to block different cytokines of the immune system such as TNF-alpha, IL12/IL23, IL17, IL17 receptor and IL-23/IL-39. Some of these biologic agents, such as secukinumab and brodalimumab, were included to treat moderate-to-severe CD, with no significant effectiveness compared to controls [34]. However, some studies evidenced that some biologics (e.g., anti-IL17 and the anti-TNF molecules) used for the treatment of psoriasis increased the risk of IBD [35,36,37,38,39], whereas the use of other biological drugs for psoriasis, such as the anti-IL23/IL39 antibodies (guselkumab, tildrakizumab, risankizumab) and anti-IL12/IL-23 inhibitor (ustekinumab), were not associated with the development of IBD [40].

The higher prevalence of psoriasis in patients with CD compared to UC [41] compelled us to focus principally on the relationship between psoriasis and Crohn’s disease associated with gut/skin microbiome dysbiosis.

## 2. Clinical Features of Psoriasis and Associated Skin and Gut Microbiome Dysbiosis

Clinical characteristics of psoriasis are different depending on the manifestation of disease. Common variants include plaque psoriasis, inverse psoriasis, guttate psoriasis, erythrodermic psoriasis and pustular psoriasis.

Plaque psoriasis is the most common form and is found in about 80% of cases [40]. It presents principally with a cutaneous involvement characterized by well-marked, erythematous and scaly lesions. They are localized on the scalp, trunk and extremities, in particular on elbows, knees and the sacral area [40,42,43].

*Inverse psoriasis*, also called intertriginous psoriasis, loses its peculiar aspect of scales and is localized in moist skin environments such as the axillary, inframammary and anogenital folds [42,44]. Other localizations are the retroauricular folds, antecubital fossae, popliteal fossae, abdomen, intergluteal cleft and perineal area [45,46].

*Guttate psoriasis* is characterized by tear-drop-shaped lesions. About 66% of the cases of guttate psoriasis are preceded by a group B streptococcal infection.

*Erythrodermic psoriasis* is an uncommon form of the disease that occurs only in 2–3% of psoriasis cases [40]. It is characterized by a gradual coalescence of plaques covering almost 75% of the body, which are red. It is considered a dermatological emergency because it can be concurrent with electrolyte alterations and desquamation that might lead to the patient’s death [40].

*Pustular psoriasis* is a form that presents with pustules containing neutrophils and is divided in two subtypes: a generalized pustular psoriasis and a localized pustular psoriasis occurring on the palms and soles [47,48,49].

Other clinical manifestations of psoriasis, besides the cutaneous involvement, include nail changes and psoriatic arthritis. 

Dysbiosis of skin microbiota has been associated with psoriasis, as reported by different studies [50,51,52,53,54]. They produced contrasting results: Gao et al. [54] found an increase in microbial diversity, while others [51,52,53] describe a decreased diversity, which is more in accordance with the literature that suggests that decreased diversity of skin microbiota is related to an altered health state [55,56]. A recent systematic review confirmed a general decrease in alpha-diversity: a higher abundance of Firmicutes and a decrease in Actinobacteria in patients with psoriasis compared to healthy controls [57]. At the genus level, *Streptococcus*, *Staphylococcus*, and *Malassezia* were associated with lesional skin, while a decrease in *Cutibacterium* was observed [57].

*Cutibacterium acnes* is a dominant bacterium in normal skin and has a protective role because it produces anti-microbial products and has immunomodulatory activity [58]. The decrease in commensal bacteria such as *Staphylococcus epidermidis* and *Cutibacterium acnes* favors the *Staphylococcus aureus* colonization, which was postulated to stimulate Th17-mediated inflammation [54]. Throat infections due to *Streptococcus pyogenes* trigger the guttate psoriasis by superantigen-induced T-cell activation [59]. The mechanism by which *Malassezia* exacerbates psoriasis is not fully elucidated. It was found that fungus components might attract polymorphonuclear leukocytes and induce cytokine release [59].

The association between dysbiosis of gut microbiota and psoriasis is less clear, with not so many studies that characterize the composition of gut microbiome in psoriasis patients. Among them, the study by Scher et al. [60] found a significant decrease (*p* < 0.05) in bacterial diversity in the gut of patients with psoriasis and psoriatic arthritis. Another study [61] found a decrease in *Akkermansia muciniphila* in psoriatic patients; this species has a protective role against inflammation and is involved in the strengthening of the intestinal barrier [61,62]. A further study showed that an elevated Firmicutes to Bacteroidetes ratio in patients with psoriasis correlated with the presence of higher inflammation [63,64]. On the contrary, the work of Codoner et al. found an increase in bacterial diversity, with an overrepresentation of *Faecalibacterium*, *Akkermansia* and *Ruminococcus* and a decrease in *Bacteroides* [65]. This last genus produces polysaccharide A and activates regulatory T cells [66], so its decrease might be related to an altered immune response. However, the increase in *Faecalibacterium* did not correspond to the increase in the species *F. praunitzii*, which was found at lower levels in both psoriasis patients and those with Crohn’s disease [65,67]. This species produces butyrate, able to inhibit the NF-kB pathway and, therefore, to block the inflammatory response (Table 1).

## 3. Clinical Features of Crohn’s Disease and Associated Gut Microbiome Dysbiosis

CD is a chronic intestinal inflammatory disease that might involve every part of the gastrointestinal tract from mouth to perianal area [68]. It might cause colitis only, ileitis only, or both [68]. Disease localization generally is stable over time.

Symptoms are heterogeneous, nonspecific and depend on the disease localization and the severity of inflammation; some patients may have symptoms that precede CD diagnosis for years [69]. The most common clinical symptoms are diarrhea, abdominal pain, fatigue, weight loss and anorexia. 

Diarrhea is due principally to decreased water absorption and increased secretion of electrolytes, while in stricturing disease it may be associated with small bowel bacterial overgrowth. The presence of systemic signs such as high-grade fever is linked to septic complications and intra-abdominal abscesses [70]. The weight loss is due to decrease oral intake and malabsorption. Bowel obstructions in patients with stricturing disease result in lack of bowel movements that gives rise to hyperactive bowel sounds, nausea and vomiting [71]. A third of patients have a perianal disease [72]; risk factors associated with this clinical manifestation are female gender and extraintestinal manifestations, while older age at diagnosis is associated with a lower risk of perianal lesions [73]. Perianal disease can manifest as skin lesions, anal canal lesions, and fistulas with or without abscesses [70]. A population-based study showed that the cumulative risk of developing fistulas was 33% at 10 years and 50% after 20 years [74]. Fistulas and abscesses are related to penetrating disease. 

About 43% of patients present extraintestinal manifestations (localized in muscular and skeletal tract and in oral, dermatological and hepatobiliary systems) that may be present even before gastrointestinal symptoms.

Gut dysbiosis, responsible for intestinal inflammation [75], has been largely studied in patients with IBD and in patients with Crohn’s disease. Most of the studies reported a decrease in Bacteroides and Firmicutes bacteria, in particular those belonging to the Clostridium species (17 strains of groups IV and XIVa, and butyricum) and an increase in Gammaproteobacteria and Actinobacteria [76,77,78,79,80,81,82]. Furthermore, a relative increase in Proteobacteria, principally *E. coli*, was observed in patients with CD mostly on mucosa-associated microbiota compared to fecal samples [83,84].

*E. coli* associated with CD is an adhesion-invasive *E. coli* strain (AIEC). The increase in bacteria able to adhere to the intestinal mucosa affects impermeability of the intestine, alters the diversity in composition of gut microbiota, and activates inflammatory responses [85,86]. In particular, these strains transit through the intestinal barrier, adhere to and invade epithelial cells, survive inside the macrophages, and stimulate the production of high levels of TNF-alpha [83,84].

Besides *E. coli* spp., *Enterococcus* spp., *Fusobacterium* spp., *Streptococcus* spp., and *Veillonella* spp. have been considered promising markers for bile duct obstruction and inflammation, associating, therefore, their presence with primary sclerosing cholangitis, one of the extraintestinal manifestations in IBD [87]. Furthermore, different studies detected a decrease in *F. prausnitzii*, *Blautia faecis*, *Roseburia inulinivorans*, *Ruminococcus torques* and *Clostridium lavalense* in CD patients compared with healthy controls [87,88]. In particular, a low number of *F. prausnitzii* was considered a marker associated with a higher risk of relapse of ileal CD after surgery [89]. It has anti-inflammatory effects because it produces butyrate and stimulates the production of anti-inflammatory cytokines such as IL-10 by mononuclear cells, blocking the production of IL12 and IFN-gamma [90]. 

In addition, the altered production of metabolites due to an imbalance of gut microbiota is associated with the pathogenesis of CD, such as a decrease in the concentration of SCFAs (short fatty acids). This reduction influences the expansion of Treg cells, which are important for the maintenance of intestinal immune homeostasis [91]. There is also a higher production of hydrogen sulfide, related to an increased number of sulfate-reducing bacteria, which induces damage to intestinal epithelial cells and promotes intestinal inflammation [92,93,94,95].

Viral and fungal dysbiosis were characterized by an expansion of *Caudovirales* viruses and *Candida* spp. detected in stool of patients with CD compared to healthy controls [96,97].

The summary of microbiome changes in psoriasis and Crohn’s disease is reported in Table 1.

## 4. The Role of Gut Microbiota Perturbation in Psoriasis and Crohn’s Disease

What is the role of gut microbiome perturbation in driving immune pathogenesis in both psoriasis and Crohn’s disease?

It is well known that microbiota has a leading role in the pathogenesis of many diseases [98]. In particular, different studies show evidence of a connection between gut and skin health state that is implied in the pathophysiology of many chronic inflammatory diseases [99,100,101,102]. The hypothesis is that the gut microbiome is responsible for the skin homeostasis and allostasis [99,103]. 

Even if the complete mechanism by which communication occurs between the gut and the skin is not fully understood, it has been reported that gut microbiota affects skin health by inducing an immune system imbalance [99]. Intestinal microbiota can determine, in fact, the direction of differentiation of naïve CD4+ T cells in effector T cells or Tregs. The balance between Tregs and effector T cells such as Th1, Th2 and Th17 is very important to maintaining an immune homeostasis. The production of some metabolites by certain bacteria present in the gut such as *Bacteroides* spp. and *Faecalibacterium prausnitzii* induces an increase in regulatory T cells and, therefore, improves anti-inflammatory responses [104], while the presence of segmented filamentous bacteria induces an increase in pro-inflammatory Th17 and Th1 cells.

So, the story might begin with an alteration of the gut microbiota: abnormal levels of commensal gut bacteria give rise to an imbalance of T cell populations, which can lead to a disruption in cutaneous homeostasis and trigger systemic inflammation in both skin and gut [105,106].

What is the relationship between gut dysbiosis and Crohn’s disease?

The cause that determines the passage from symbiosis to dysbiosis is not well understood, even if a genetic predisposition, diet and/or environmental factors have been postulated [107]. 

In an inflammatory disease such as Crohn’s disease, genetic susceptibility linked to polymorphisms in NOD2 and NF-kB signaling pathway genes gives rise to epithelial barrier dysfunction. At this point, the luminal contents gain access to the lamina propria, inducing dendritic cells to activate inflammatory T cells such as Th1, Th2 and Th17. They produce inflammatory cytokines such as IFN-gamma, TNF-alpha, IL-4, Il-6, IL-21 and IL-22. Furthermore, macrophages produce IL-12 and IL-23 [69] (Figure 1). Moreover, dysbiosis alters the production of different metabolites in the intestine, increasing, for example, the amount of hydrogen sulfide, which is correlated to mucosal inflammation because it inhibits butyrate production [108]. Next, enteric anaerobe digesting carnitine and phosphatidylcoline introduced with diet produce trimethylamine-N-oxide, which contributes to mucosal inflammation [109]. Conversely, a reduction in metabolites with a protective role was observed, such as butyrate and propionate, which normally stimulate the expansion of regulatory T cells [110]. In addition, oxidative stress produced during gut dysbiosis and due to an increase in oxidative free radicals is a mechanism that predisposes patients to the pathogenesis [111]. It has been proposed that alteration of gut microbiota increases the production of nitric oxide and nitric oxide synthases that affect the DNA repair system and induce cell membrane dysfunction [111].

Is gut dysbiosis and consequent immune alteration in Crohn’s disease linked to psoriasis pathology, or vice versa?

One hypothesis is that Th1 cytokines hyperproduction during inflammatory intestinal disease might be strongly linked with psoriasis, with a central role attributed to a class of memory T cells characterized by the presence of the cutaneous lymphocyte antigen (CLA) on their surface and responsible for skin-homing [112,113] (Figure 1).

When exposed to a cytokine inflammatory environment and/or to skin-associated antigens and/or to microbial antigens, these T cells, which circulate between blood and skin, respond by activating Th17 and Th9 cells, which produce IL17 and IL9, respectively [114]—important mediators of psoriasis pathogenesis [115]. IL-15 produced by damaged keratinocytes and IL-23 produced by dendritic cells expand further the Th17 population, contributing to the inflammatory loop underlying psoriasis lesions [116]. The inflammatory cytokines released, such as IL17A, IL17 F, TNF-alpha, IFN-alpha and IL-22, are the trigger of keratinocyte hyperproliferation and skin thickening that characterize psoriatic lesions [117,118] (Figure 1).

Another hypothesis postulated to explain the mechanism by which gut dysbiosis can induce psoriasis pathogenesis involves increased permeability that allows intestinal bacteria and/or their metabolites to reach blood circulation and skin [99,119]. In support of this theory, DNA of intestinal bacteria has been found in the bloodstream of patients with chronic skin diseases, contributing to inflammatory response [65,99]. Analogously, metabolites of gut bacteria may gain access to blood and accumulate in the skin, reducing keratin synthesis and epidermal differentiation [99,120]. Dysbiosis of gut microbiota may contribute to the increased intestinal permeability, improving the bacterial translocation, which may trigger the inflammatory response [119,120,121,122]. Translocation may occur through not only the intestinal epithelial barrier, but also dendritic and M cells [123]. These bacteria may not be replicating, but nevertheless are able to maintain a low state of chronic inflammation in the patient. They occasionally discharge parts of their cell walls, such as lipopolysaccharide (LPS) and/or teichoic acid (LTA), which activate innate and adaptive immunity by inducing the production of inflammatory cytokines and chemokines and contributing to a sustained, low-grade, chronic inflammatory status in the host organism, which in turn induces the formation of psoriatic plaques [124,125].

Some metabolites, such as propionate, acetate and butyrate produced by nutrient digestion in the gut, play an important role in influencing the composition of skin microbiome. The absence of these short-chain fatty acids might be responsible for an altered cutaneous commensal microbiota.

The innate immune system presents in the skin by using different receptors such as Toll-like receptors (TLRs), pattern recognition receptors (PRRs), proteoglycan recognition proteins (PRGRPs), and is able to recognize altered cutaneous microbiota, microbes or microbial antigens translocated from the gut [126] and to launch altered innate and adaptive immune responses [58] (Figure 1).

## 5. Conclusions

Establishing from these studies the causality between Crohn’s disease and psoriasis after gut microbiota dysbiosis and altered immune responses is still a puzzle. We could say that we are facing an example of the age-old “chicken or the egg” dilemma.

Psoriasis is linked to systemic inflammation, which may cause changes in the gut mucosa and, as a result, influence the intestinal immune homeostasis. On the other hand, the disequilibrium of the gut microbiome, influenced by diet, environment and genetic predisposition, triggers a proinflammatory environment in the gut responsible for chronic intestinal inflammation (Crohn’s disease) and able to generate systemic inflammation, which can lead to skin inflammation. Future research is needed to determine the exact mechanism involved.

However, if these diseases are mediated by an altered gut microbiota composition, its modulation and its communication with the host’s regional and systemic immune system represent an important opportunity to improve their outcome.

Considering that both psoriasis and Crohn’s disease are linked to a sustained immune dysfunction generated by a gut microbiota dysbiosis that can negatively influence immunological homeostasis, restoring normal microbiota composition is critical for disease therapy, even when genetic, epigenetic, and environmental risk factors are taken into account.

## Figures and Tables

**Figure 1 vaccines-10-00244-f001:**
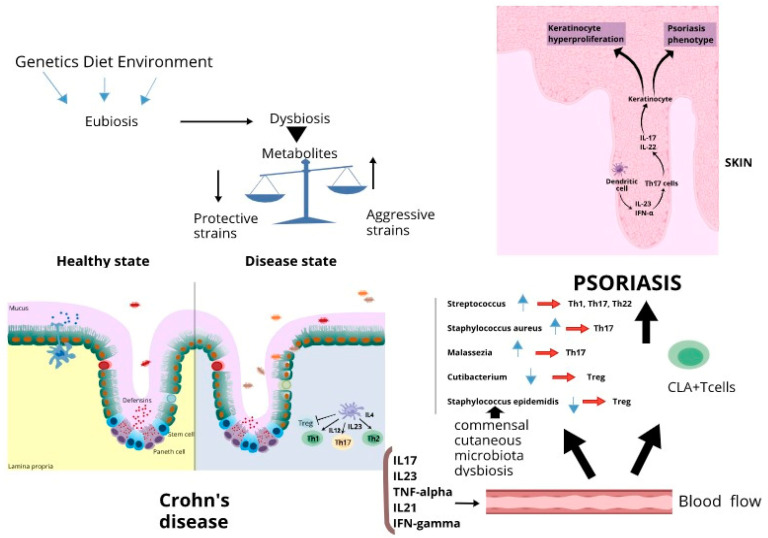
Link between gut microbiota dysbiosis, immune response dysregulation, Crohn’s disease and psoriasis.

**Table 1 vaccines-10-00244-t001:** Microbiome changes in psoriasis and Crohn’s disease according to different studies.

Crohn’s Disease				Psoriasis			
Increased *	Reference	Decreased ^§^	Reference	Increased *	Reference	Decreased ^§^	Reference
*Escherichia coli*	[83,84]	*Bacteroides*	[76,77,78,79,80,81,82]	*Streptococcus pyogenes*	[59]	*Staphylococcus epidermidis*	[54]
*Enterococcus* spp.	[87]	*Clostridium clusters XIVa and IV*	[76,77,78,79,80,81,82]	*Staphylococcus aureus*	[54]	*Cutibacterium acnes*	[54]
*Fusobacterium* spp.	[87]	*Firmicutes prausnitzii*	[87,88]	*Malassezia* spp.	[59]	*Akkermansia muciniphila*	[86,87]
*Streptococcus* spp.	[87]	*Blautia faecis*	[87,88]			*Bacteroides*	[65]
*Veillonella* spp.	[87]	*Roseburia inulinivorans*	[87,88]			*Firmicutes praunitzii*	[65,67]
		*Ruminococcus torques*	[87,88]				
		*Clostridium lavalense*	[87,88]				

* Species associated with inflammatory effects; ^§^ species associated with protective immunomodulatory effects.

## Data Availability

Not applicable.

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
