# Peer review of "The Gut Microbiome in Psoriasis and Crohn’s Disease: Is Its Perturbation a Common Denominator for Their Pathogenesis?"

_vaccines, 2022, doi:10.3390/vaccines10020244_

Round 1

Reviewer 1 Report

In this review, Drs. De Francesco and Caruso provide an interesting summary about the current status of research in the microbiome role in two specific autoimmune diseases, IBD (mainly Crohn disease) and psoriasis. I have just several minor comments.

Ln 37. At the beginning of the introduction, I think it should be explained better what the link between IBD, UC, and CD is; subsequently, the terms IBD, UC, and CD can be used with a clear idea in mind of their relationship.

Ln 62: CD? Nor IBD? Genetic results refer to both diseases, as indicated in ln 64.

In general, I think it is not perfectly clear why several points are highlighted for IBD, others for CD only: for example, in lines 69-73, 4 loci have been declared to be associated with IBD first, then with psoriasis and CD only (while they are pleiotropic and associated also with UC). An effort should be done to clarify better in the introduction why the authors concentrate the rest of the paper mainly on CD and psoriasis comparison.

The information about clinical aspects is fine and the paragraph about the role of skin and gut microbiota on the diseases collects the more recent data published and could be useful for a general audience. 

Author Response

In this review, Drs. De Francesco and Caruso provide an interesting summary about the current status of research in the microbiome role in two specific autoimmune diseases, IBD (mainly Crohn disease) and psoriasis. I have just several minor comments.

 Ln 37. At the beginning of the introduction, I think it should be explained better what the link between IBD, UC, and CD is; subsequently, the terms IBD, UC, and CD can be used with a clear idea in mind of their relationship.

Answer: We agree with the reviewer and we have added a brief sentence at the beginning of the Introduction section

 Ln 62: CD? Nor IBD? Genetic results refer to both diseases, as indicated in ln 64. In general, I think it is not perfectly clear why several points are highlighted for IBD, others for CD only: for example, in lines 69-73, 4 loci have been declared to be associated with IBD first, then with psoriasis and CD only (while they are pleiotropic and associated also with UC).

Answer: We have corrected the mistake

 An effort should be done to clarify better in the introduction why the authors concentrate the rest of the paper mainly on CD and psoriasis comparison. The information about clinical aspects is fine and the paragraph about the role of skin and gut microbiota on the diseases collects the more recent data published and could be useful for a general audience

Answer: We specified as suggested by the reviewer in the introduction that we decided to focus our review principally on CD and psoriasis because the percentage of association was higher for these diseases than UC.

Reviewer 2 Report

This review examines the gut microbiome in an effort to link Crohn's disease and psoriasis. I have seen this mentioned in genetic studies. The review begins with background on factors that show associations between the diseases.  While this has been done elsewhere, this part of the review is the best written; brief and to the point.  It make a case for examination of the gut microbiota.  The review bogs down on the clinical presentations sections.  The sections are too long and bring up to much detail that does not add to the points raised in section 4. These 2 sections need to be trimmed down so that they add really needed information to set up section 4.

The heart of the review is section 4, which is not well written.  The points of discussion are there.  The way the points are discussed leaves me no real strong impression of the authors take on the interaction of gut microbiota and psoriasis.  It needs reorganization; a point is made about CD and immediately connected to psoriasis, then move on to the next point.  Also, there is one point one which I disagree. On line 266, the authors talk about gut microbiota binding directly with epithelial cells.  I have seen pathogens do this in my own samples (electron microscope pictures of colon and ileum samples of mouse models of IBD), but only rarely and never did I see this going on with commensals.  I am more inclined to the 2 layers of mucin idea, with an inner sterile layer (Hannson, G, 2912 Current Opinion in Microbiology, 15 (1) pp 57-62).  This still allows for bacterial products to have access to the epithelium and of course pathogens can penetrate.  

Author Response

This review examines the gut microbiome in an effort to link Crohn's disease and psoriasis. I have seen this mentioned in genetic studies. The review begins with background on factors that show associations between the diseases.  While this has been done elsewhere, this part of the review is the best written; brief and to the point.  It make a case for examination of the gut microbiota. 

Answer: This part has been unchanged

The review bogs down on the clinical presentations sections.  The sections are too long and bring up to much detail that does not add to the points raised in section 4. These 2 sections need to be trimmed down so that they add really needed information to set up section 4.

Answer: These two sections have been shortened and associated to the part that deals about gut and skin dysbiosis

The heart of the review is section 4, which is not well written.  The points of discussion are there.  The way the points are discussed leaves me no real strong impression of the authors take on the interaction of gut microbiota and psoriasis.  It needs reorganization; a point is made about CD and immediately connected to psoriasis, then move on to the next point.  Also, there is one point one which I disagree. On line 266, the authors talk about gut microbiota binding directly with epithelial cells.  I have seen pathogens do this in my own samples (electron microscope pictures of colon and ileum samples of mouse models of IBD), but only rarely and never did I see this going on with commensals.  I am more inclined to the 2 layers of mucin idea, with an inner sterile layer (Hannson, G, 2912 Current Opinion in Microbiology, 15 (1) pp 57-62).  This still allows for bacterial products to have access to the epithelium and of course pathogens can penetrate

Answer: We agree with the referee and we reorganized all the section 4 trying to better explain the link between the gut dysbiosis, the following immune alteration, the inflammation in Crohn’s disease and the possible relationship with psoriasis. We have deleted the point evidenced by the reviewer.

Reviewer 3 Report

Authors discuss the relationship between gut dysbiosis and IBD or psoriasis. Although not new, the  subject is interesting to the readers and the data on the topic evolve over recent years.

Major concerns:

  1. The title is not fully compatibile withe the text. In fact, authors describe IBD not CD only. I suggest changing the title and the subtitles (line 199 and 260), accordingly.
  2. The abstract: according to the title of the manuscript, the microbiota (lines 24 to 27) should be mentioned before genetic predispositions.
  3. To large volume of the manuscript is covered by clinical presentation of PS and IBD (paragraph 2 and 3. Whereas, neither symptoms nor disease forms are discussed in relation to microbiome there. I suggest removing these paragraphs, as they do not add value to the paper. Instead, some clinical information should be used and discussed in the paragraph 4.
  4. Paragraph 4.2: The table summarizing the relationship between bacteria and clinical findings would make the text easier and more interesting to the readers.

Minor comment:

  1. Please, verify the risk of 1,6 in IBD or in CU - the data presented in line 13 and 39 are not the same.
  2. Please, verify meaning of the sentence in line 20
  3. Please, verify the prevalence and incidence of PS in general population (line 43).
  4. Please, be more specific in line 51 (some other autoimmune...).
  5. Please, spell check the manuscript.

Author Response

Authors discuss the relationship between gut dysbiosis and IBD or psoriasis. Although not new, the  subject is interesting to the readers and the data on the topic evolve over recent years.

Major concerns:

  1. The title is not fully compatibile withe the text. In fact, authors describe IBD not CD only. I suggest changing the title and the subtitles (line 199 and 260), accordingly.

Answer: We have specified in the introduction as suggested also by the reviewer 3 that even if there are some common characteristics between both UC and CD and psoriasis, we tried to focus the review only on CD and psoriasis

  1. The abstract: according to the title of the manuscript, the microbiota (lines 24 to 27) should be mentioned before genetic predispositions.

Answer: According to the reviewer’s suggestion, we have made the change

  1. To large volume of the manuscript is covered by clinical presentation of PS and IBD (paragraph 2 and 3. Whereas, neither symptoms nor disease forms are discussed in relation to microbiome there. I suggest removing these paragraphs, as they do not add value to the paper. Instead, some clinical information should be used and discussed in the paragraph 4.

Answer: These two sections have been shortened and associated to the part that deals about gut and skin dysbiosis. They were not completely removed as suggested by the reviewer to satisfy the observation done by the third reviewer.

  1. Paragraph 4.2: The table summarizing the relationship between bacteria and clinical findings would make the text easier and more interesting to the readers.

Answer: According to the reviewer’s suggestion, a table has been inserted

Minor comment:

  1. Please, verify the risk of 1,6 in IBD or in CU - the data presented in line 13 and 39 are not the same.

Answer: the mistake has been corrected

  1. Please, verify meaning of the sentence in line 20

Answer: We have changed the sentence

  1. Please, verify the prevalence and incidence of PS in general population (line 43).

Answer: We have reported the percentage (10%) inserted in a recent paper (Nehring P et al., Pharmaceutical Medicine, 2020), but looking at other papers we have observed that the percentage is about 2-3% and therefore we have corrected it

  1. Please, be more specific in line 51 (some other autoimmune...).

Answer: We have added the autoimmune diseases

  1. Please, spell check the manuscript.

Answer: The manuscript has been spell checked

Round 2

Reviewer 2 Report

This review is much closer to being publishable.  The last section being the most revised has grammar problems that should be corrected.

Line 162 delete "So far; separate cause/colitis

Line 183-responsible "for"

Line 202-because "it"

Line 209- sulfide is the reduction product

Lines 219 and 239- the interrogative should be "What"

Line 240- The cause that determine"s"

Line 259-"or vice versa"

Author Response

All the mistakes indicated by the reviewer have been corrected.

Thanks for giving us this opportunity

Reviewer 3 Report

Revision accepted.

Author Response

The reviewer has accepted our revision